# The Drosophila Sp8 transcription factor Buttonhead prevents premature differentiation of intermediate neural progenitors

Yonggang Xie[1], Xiaosu Li[1], Xian Zhang[1], Shaolin Mei[1], Hongyu Li[1], Andreacarola Urso[2], Sijun Zhu[1]*

[1]Department of Neuroscience and Physiology, State University of New York Upstate Medical University, Syracuse, United States; [2]Department of Biology, Syracuse University, Syracuse, United States

**Abstract** Intermediate neural progenitor cells (INPs) need to avoid differentiation and cell cycle exit while maintaining restricted developmental potential, but mechanisms preventing differentiation and cell cycle exit of INPs are not well understood. In this study, we report that the *Drosophila* homolog of mammalian Sp8 transcription factor Buttonhead (Btd) prevents premature differentiation and cell cycle exit of INPs in *Drosophila* larval type II neuroblast (NB) lineages. We show that the loss of Btd leads to elimination of mature INPs due to premature differentiation of INPs into terminally dividing ganglion mother cells. We provide evidence to demonstrate that Btd prevents the premature differentiation by suppressing the expression of the homeodomain protein Prospero in immature INPs. We further show that Btd functions cooperatively with the Ets transcription factor Pointed P1 to promote the generation of INPs. Thus, our work reveals a critical mechanism that prevents premature differentiation and cell cycle exit of *Drosophila* INPs.

*For correspondence: zhus@ upstate.edu

**Competing interests:** The authors declare that no competing interests exist.

**Reviewing editor**: Marianne E Bronner, California Institute of Technology, United States

## Introduction

Intermediate neural progenitor cells (INPs) play a critical role in increasing the brain size and complexity. Transient amplification of INPs dramatically boosts the neural output from neural stem cells (NSCs) (*Kriegstein and Alvarez-Buylla, 2009*; *Florio and Huttner, 2014*). Recent studies in developing human brains as well as other mammalian brains suggest that an expansion of the number of transiently amplifying INPs, the outer sub-ventricular zone radial glia-like cells (oRGs), likely contributes to the increased cortical size and complexity in humans and other gyrencephalic animals (*Fietz et al., 2010*; *Hansen et al., 2010*; *Lui et al., 2011*; *Wang et al., 2011*). On the other hand, accumulating body of evidence suggests that brain tumors could originate from dedifferentiation and unrestricted proliferation of INPs (*Holland et al., 2000*; *Dai et al., 2001*; *Walton et al., 2009*; *Persson et al., 2010*; *Zong et al., 2012*). Therefore, it is fundamentally important to understand how the generation and proliferation of INPs are regulated.

The recently discovered type II neuroblasts (NBs, the *Drosophila* NSCs) in developing *Drosophila* larval brains provide an excellent model system for studying mechanisms regulating the generation and proliferation of INPs (*Bello et al., 2008*; *Boone and Doe, 2008*; *Bowman et al., 2008*). There are 8 type II NBs in each brain lobe. Like mammalian NSCs, *Drosophila* type II NBs produce neurons and glia indirectly by generating INPs. Individual INPs undergo 4–6 rounds of asymmetric divisions to produce a new INP to self-renew and a ganglion mother cell (GMC), which divides terminally to produce neurons and/or glia (*Bayraktar et al., 2010*; *Viktorin et al., 2011*; *Yang et al., 2013*). Meanwhile,

**eLife digest** Whereas the majority of cells in the brain are unable to divide to produce new cells, neural stem cells can divide numerous times and have the potential to become many different types of brain cells. However, in between these two extremes there is another group of cells called neural progenitors. These cells can give rise to multiple types of neurons but, in contrast to stem cells, they can undergo only a limited number of divisions.

Many of the molecular mechanisms by which stem cells give rise to progenitors are similar in mammals and in the fruit fly *Drosophila*. In the brains of fruit fly larvae, neural stem cells called neuroblasts give rise to 'intermediate neural progenitors', each of which can divide between four and six times. Every division generates a replacement intermediate progenitor and a cell called a GMC, which divides one last time to produce two brain cells.

Intermediate progenitors must be tightly regulated to ensure that they undergo an appropriate number of divisions: too few divisions will result in a shortage of cells, disrupting brain development, whereas too many divisions will result in the formation of tumors. Now, using *Drosophila* brains in the laboratory, Xie et al.—and, independently, Komori et al.—have shown that a protein called 'Buttonhead' is responsible for maintaining this balance.

Xie et al. show that deletion of the gene for Buttonhead gene caused the progenitor cells to become GMCs before they had undergone the correct number of divisions. Further experiments revealed that Buttonhead prevents this problem by suppressing a protein called Prospero.

The mammalian equivalent of Buttonhead—a protein called Sp8—can substitute for Buttonhead in *Drosophila* neural progenitors, suggesting that the observed mechanisms may also apply to mammals. Further work is required to test this possibility directly and to examine the involvement of Sp8 in brain development and tumor formation.

individual INPs produce distinct types of neurons by sequentially expressing a set of distinct transcription factors to specify the identity of their progeny (*Bayraktar and Doe, 2013*; *Wang et al., 2014*). Through self-renewing divisions, INPs not only amplify the number but also increase the diversity of neural progeny generated from type II NBs. Therefore, the neurogenesis pattern in type II NB lineages is remarkably similar to that in mammalian brains and the *Drosophila* INPs are functionally analogous to mammalian INPs, particularly oRGs.

The generation of INPs in type II NB lineages involves multiple steps (*Bello et al., 2008*; *Boone and Doe, 2008*; *Bowman et al., 2008*). Newly generated INPs are immature and do not express any NB markers, such as the proneural protein Asense (Ase) or the bHLH protein Deadpan (Dpn), except for Miranda (Mira). The Ase$^-$ immature INPs first turn on the expression of Ase to become Ase$^+$ immature INPs. Ase$^+$ immature INPs then further differentiate to become mature INPs, which express both Ase and Dpn. INPs do not divide until they are fully mature. The maturation of INPs requires Numb, the NHL family protein Brain tumor (Brat), the transcription factor Earmuff (Erm), as well as the BAP and Histone deacetylase 3 (HDAC3) chromatin remodeling complexes (*Bowman et al., 2008*; *Weng et al., 2010*; *Eroglu et al., 2014*; *Koe et al., 2014*). Both Numb and Brat are segregated into Ase$^-$ immature INPs during the division of type II NBs to prevent them from dedifferentiating into NB fate, but they function through independent pathways. Numb inhibits Notch activity in Ase$^-$ immature INPs, whereas Brat likely antagonizes the activity of the EGR family transcription factor Klumpfuss (Klu) and Armadillo/β-Catenin in Ase$^-$ immature INPs (*Bowman et al., 2008*; *Komori et al., 2014*). Erm functions together with BAP and HDAC3 chromatin remodeling complexes after Brat and Numb to further restrict the developmental potential of INPs by attenuating the response of INPs to self-renewal factors such as Klu and Dpn (*Janssens et al., 2014*; *Koe et al., 2014*). In addition, the BAP chromatin remodeling complex limits the self-renewal of INPs by activating the expression of Prdm protein Hamlet (*Eroglu et al., 2014*). In the absence of Numb, Brat, Erm, or chromatin remodeling complexes, INPs dedifferentiate into type II NBs and initiate tumorigenic overproliferation (*Bowman et al., 2008*; *Weng et al., 2010*; *Eroglu et al., 2014*; *Koe et al., 2014*). Therefore, these proteins are critical to prevent dedifferentiation of INPs.

However, despite the significant progress on elucidating mechanisms that promote maturation and prevent dedifferentiation of INPs in the past few years, much less is known about why only type II NBs

produce self-renewing INPs but not the type I NBs, which produce neurons by generating terminally dividing GMCs. One major difference between INPs and GMCs is that INPs divide to self-renew whereas GMCs divide terminally (*Bello et al., 2008*; *Boone and Doe, 2008*; *Bowman et al., 2008*; *Yang et al., 2013*). Therefore, in addition to avoiding dedifferentiation and unrestricted tumorigenic overproliferation, INPs need to overcome another challenge–to avoid over-differentiation and cell cycle exit–in order to maintain their progenitor state and self-renewal while they differentiate to mature and undergo self-renewing divisions. Type II NBs and newly born Ase⁻ immature INPs differ from type I NBs and GMCs by the lack of the expression of Ase and the homeodomain protein Prospero (Pros) (*Bello et al., 2008*; *Boone and Doe, 2008*; *Bowman et al., 2008*). In type I NB lineages, Pros is expressed in the cytoplasm in the NBs and translocates to the nucleus in GMCs to promote differentiation and cell cycle exit by inhibiting NB self-renewing genes and activating neural differentiation genes (*Li and Vaessin, 2000*; *Choksi et al., 2006*). In type II NB lineages, Pros is not expressed in the NB or Ase⁻ immature INPs. In Ase⁺ immature INPs and mature INPs, Pros is expressed at low levels in the cytoplasm (*Bello et al., 2008*; *Boone and Doe, 2008*; *Bowman et al., 2008*). It has been demonstrated that the lack of Ase and Pros in type II NBs and Ase⁻ immature INPs is essential for the generation of self-renewing INPs in type II NB lineages. Forced expression of Ase or Pros in type II NBs and their progeny is sufficient to eliminate INPs although removing Ase or Pros in type I NBs does not change the identity of type I NBs or induce the generation of INPs (*Bowman et al., 2008*; *Bayraktar et al., 2010*; *Zhu et al., 2012*). Our recent studies demonstrated that the Ets family transcription factor Pointed P1 (PntP1) suppresses Ase in type II NBs and is required for the generation of INPs (*Zhu et al., 2011*). However, mechanisms that prevent premature differentiation of INPs and/or inhibit Pros expression in type II NBs and immature INPs are not known.

In this study, we investigate the role of the Sp family transcription factor Buttonhead (Btd) in type II NB lineage development. Btd is a homolog of mammalian Sp8 (*Treichel et al., 2003*; *Estella and Mann, 2010*). In developing mammalian brains, Sp8 is expressed in neural progenitor cells to regulate forebrain patterning and interneuron development (*Griesel et al., 2006*; *Waclaw et al., 2006*; *Sahara et al., 2007*; *Li et al., 2011*). In *Drosophila* embryos, Btd is required for the formation of specific head segments and NB formation (*Wimmer et al., 1993*; *Younossi-Hartenstein et al., 1997*). In addition, Sp8/Btd also promotes the growth of limbs and other appendages (*Estella et al., 2003*; *Treichel et al., 2003*; *Kawakami et al., 2004*; *Estella and Mann, 2010*). In this study, we report that Btd is expressed in type II NB lineages to prevent premature differentiation of INPs into GMCs by suppressing Pros expression in immature INPs. We also demonstrate that PntP1 and Btd function cooperatively to specify type II NB lineages and promote the generation of INPs.

## Results

### Loss of Btd results in a complete elimination of mature INPs in type II NB lineages

Our recent studies demonstrated that the Ets family transcription factor PntP1 is specifically expressed in type II NB lineages to promote the generation of INPs. However, although forced expression of PntP1 suppress Ase expression in nearly all type I NB lineages, it induces the generation of INP-like cells only in a subset of type I NBs (*Zhu et al., 2011*). Therefore, it is likely that other protein(s) may function together with PntP1 to specify type II NB lineages and promote the generation of INPs. A recent functional genomic study showed that in addition to PntP1, there are other nine genes that are highly expressed in brain tumors derived from type II NB lineages (*Carney et al., 2012*). We wondered whether any of these genes could function together with PntP1 to promote INP generation. To test this idea, we first examined how knockdown of these genes would affect INP generation in type II NB lineages. A normal type II NB lineage contains 2–3 Ase⁻ immature INPs, 2–3 Ase⁺ immature INP, and about 20–30 (26.9 ± 4.1, mean ± SD) Ase⁺ Dpn⁺ mature INPs (*Figure 1A–A',C–C',G*). Interestingly, RNAi knockdown of Btd using the type II NB lineage-specific *pntP1-GAL4* (named as *GAL4¹⁴⁻⁹⁴* previously) (*Zhu et al., 2011*) as a driver led to a complete elimination of mature INPs in about 50% of type II NB lineages (*Figure 1B–B',G–H*). Instead, only a few (3.7 ± 1.2) Ase⁺ Dpn⁻ cells were observed next to the Ase⁻ immature INPs (*Figure 1B–B'*). However, type II NBs remain Ase⁻ as normal type II NBs (*Figure 1B–B'*), suggesting that the identity of the type II NBs was not affected by Btd RNAi knockdown.

To confirm that the loss of INPs indeed results from the knockdown of Btd rather than off-target effects of *UAS-Btd RNAi*, we generated *btd* mutant type II NB clones using two loss-of-function alleles,

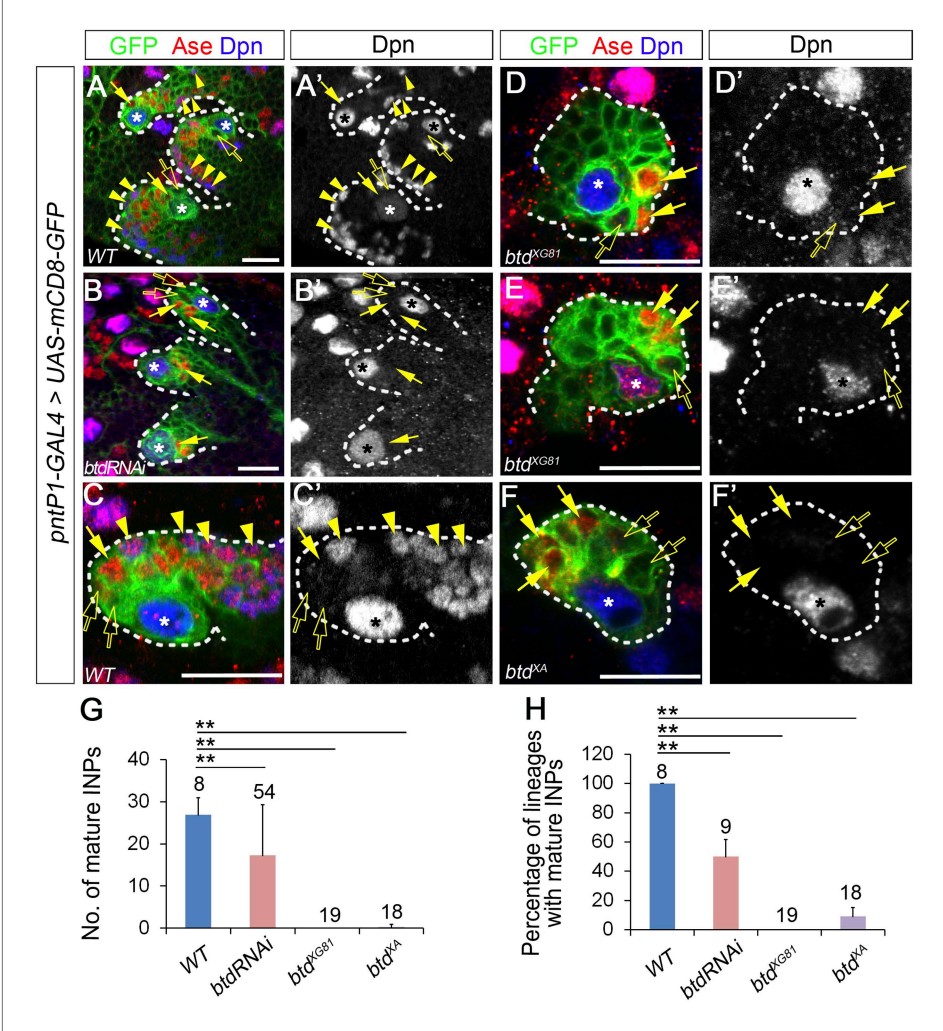

**Figure 1**. Loss of Btd eliminates mature INPs in type II NB lineages. (**A–A'**, **C–C'**) Wild-type type II NB lineages in a third instar larval brain. mCD8-GFP driven by pntP1-GAL4 labels all type II NB lineages (**A–A'**) or a single type II NB clone (**C–C'**). Ase⁻ immature INPs, Ase⁺ immature INPs, and mature INPs are indicated by open arrows, solid arrows, and arrowheads, respectively. (**B–B'**, **D–F'**) Btd RNAi knockdown type II NB lineages (**B–B'**) or type II NB clones homozygous mutant for *btd^XG81* (**D–E'**) or *btd^XA* (**F–F'**) in 3rd instar larval brains produce Ase⁻ immature INPs (open arrows) and a few Ase⁺ daughter cells (arrows) but no mature INPs. Only 3 out of total 8 type II NB lineages are shown in (**A–A'**) and (**B–B'**). In this and all other figures, asterisks indicate type II NBs and scale bars equal to 20 μm. Dpn staining alone shows the NB and mature INPs. (**G–H**) Quantifications of the number of mature INPs (**G**) and the percentage of type II NB lineages with mature INPs (**H**) in the wild type, Btd RNAi knockdown, and btd mutant type II NB lineages. The numbers on top of each bar are the numbers of type II NB lineages analyzed except for the numbers for the wt and btd RNAi in (**H**), which are the number of brain lobes examined. The mean and stdev for *btd^XG81* and *btd^XA* in (**H**) are calculated by bootstrapping. **p < 0.01, *p < 0.05 (Student *t* test).

The following figure supplement is available for figure 1:

**Figure supplement 1**. Expression of mouse Sp8 (mSp8).

---

*btd^XA* and *btd^XG81* (**Wimmer et al., 1993**; **Estella and Mann, 2010**). Consistent with the Btd RNAi knockdown, all *btd^XG81* mutant and 90% of *btd^XA* mutant type II NB clones failed to generate any mature INPs except for 4–6 Ase⁺ Dpn⁻ cells (**Figure 1D–F',G–H**). Moreover, about 40% of btd mutant type II NBs ectopically express Ase, making them appear as type I NB lineages (**Figure 1E**). The loss of INPs resulting from the Btd RNAi knockdown and *btd* loss-of-function mutations suggests that Btd is required for the generation of INPs. Remarkably, the loss of INPs in btd mutant clones can

be similarly rescued by the expression of mouse Sp8 or *Drosophila* Btd (*Figure 1—figure supplement 1*), suggesting that mammalian Sp8 could have a conserved role in promoting the generation of transient amplifying INPs.

Since the loss of mature INPs occurred even when Ase was not ectopically expressed in *btd* mutant type II NBs, the loss of INPs is not primarily due to the ectopic Ase expression or transformation of type II NBs into type I NBs. Therefore, we first focused our phenotypic analyses on lineages without the ectopic Ase expression in the NB. We also used the *btd*$^{XG81}$ allele for further mutant phenotypic analyses below, given that *btd*$^{XG81}$ shows slightly stronger phenotypes than *btd*$^{XA}$.

## Ase$^-$ immature INPs differentiate into Ase$^+$ immature INPs normally in the absence of Btd

Why does the loss of Btd lead to the elimination of mature INPs? When mature INPs are eliminated in the absence of Btd, the type II NBs without the ectopic Ase expression still produce Ase$^-$ immature INPs and a few Ase$^+$ Dpn$^-$ daughter cells. In normal type II NB lineages, Ase$^+$ Dpn$^-$ cells can be either Ase$^+$ immature INPs or GMCs. Therefore, three possible scenarios could happen when mature INPs are eliminated in the absence of Btd: 1) Ase$^-$ immature INPs differentiate into GMCs instead of Ase$^+$ immature INPs; 2) Ase$^-$ immature INPs differentiate into Ase$^+$ immature INPs, which then directly differentiate into neurons/glia without further dividing; 3) Ase$^-$ immature INPs differentiate into Ase$^+$ immature INPs, which in turn differentiate into terminally dividing GMCs. To distinguish these possibilities, we first wanted to determine if Ase$^-$ immature INPs still differentiate into Ase$^+$ immature INP in the absence of Btd by examining the expression of INP specific marker R9D11-CD4-*tdTomato* and progenitor marker Miranda (Mira) in the Ase$^+$ cells next to the Ase$^-$ immature INPs. R9D11-CD4-*tdTomato* utilizes a DNA fragment R9D11 from the *erm* promoter to drive the expression of CD4-tdTomato (*Han et al., 2011*). In normal type II NB lineages, R9D11-CD4-*tdTomato* is first turned on in Ase$^+$ immature INPs and becomes stronger as INPs mature (*Figure 2A–A'*), which is similar to R9D11-mCD8-GFP (*Zhu et al., 2011*). Mira is expressed in all NBs as well as INPs but not (or very weakly) in GMCs (*Figure 2——figure supplement 1*). In Btd RNAi knockdown type II NB lineages without mature INPs, we found that R9D11-CD4-*tdTomato* was expressed in Ase$^+$ daughter cells next to the Ase$^-$ immature INPs but its overall expression was much weaker than that in normal type II NB lineages (*Figure 2B–B',I*). Consistently, Mira is also expressed in those Ase$^+$ cells next to the Ase$^-$ immature INPs in the Btd RNAi knockdown type II NB lineages (*Figure 2—figure supplement 1*). The expression of R9D11-CD4-*tdTomato* and Mira suggests that Ase$^-$ immature INPs still differentiate into Ase$^+$ immature INP in the absence of Btd as in wild-type type II NB lineages (*Figure 2J*).

## Loss of Btd leads to ectopic expression of Pros in immature INPs and premature differentiation of Ase$^+$ immature INPs into GMCs

Next we asked if Ase$^+$ immature INPs differentiate into neurons/glia directly or GMCs in the absence of Btd. GMCs express both Ase and nuclear Pros and divide terminally but do not form a Mira crescent while dividing. If Ase$^+$ immature INPs directly differentiate into neurons/glia, then all the Ase$^+$ daughter cells should be Ase$^+$ immature INPs and none of them should be dividing. In contrast, if Ase$^+$ immature INPs differentiate into GMCs, then some Ase$^+$ daughter cells will become mitotically active and express nuclear Pros but will not form a Mira crescent at the metaphase or telophase. Immunostaining with the mitotic marker phospho-histone 3 (pH3) showed that unlike Ase$^+$ immature INPs, which never become pH3-positive (*Figure 2C–C'*), some Ase$^+$ daughter cells generated in both Btd RNAi knockdown type II NB lineages without mature INPs (*Figure 2D–D'*) and *btd* mutant type II NB lineages became mitotically active (*Figure 2E–E'*). However, the pH3 positive cells were always the furthest from the Ase$^-$ immature INPs among the Ase$^+$ daughter cells (*Figure 2D–D'*), suggesting that the Ase$^+$ daughter cells divide terminally like GMCs. Consistently, unlike in mature INPs, which form a Mira crescent at metaphase (*Figure 2—figure supplement 1*), we did not observe any Mira crescents in the Ase$^+$ daughter cells at metaphase (*Figure 2—figure supplement 1*). The terminal division and the lack of the Mira crescent strongly argue that Ase$^+$ immature INPs differentiate into GMCs in the absence of Btd.

To further confirm that late immature INPs differentiate into GMCs in the absence of Btd, we then examined the expression of nuclear Pros in the Ase$^+$ daughter cells. Nuclear Pros is a cell fate determinant of GMCs. In normal type II NBs lineages, Pros is expressed in the cytoplasm of Ase$^+$ immature INPs and mature INPs and in the nucleus of GMCs and post-mitotic neurons, but not in type II NBs or Ase$^-$ immature INPs (*Figure 2F–F'*). If Ase$^+$ immature INPs differentiate into GMCs in the absence of

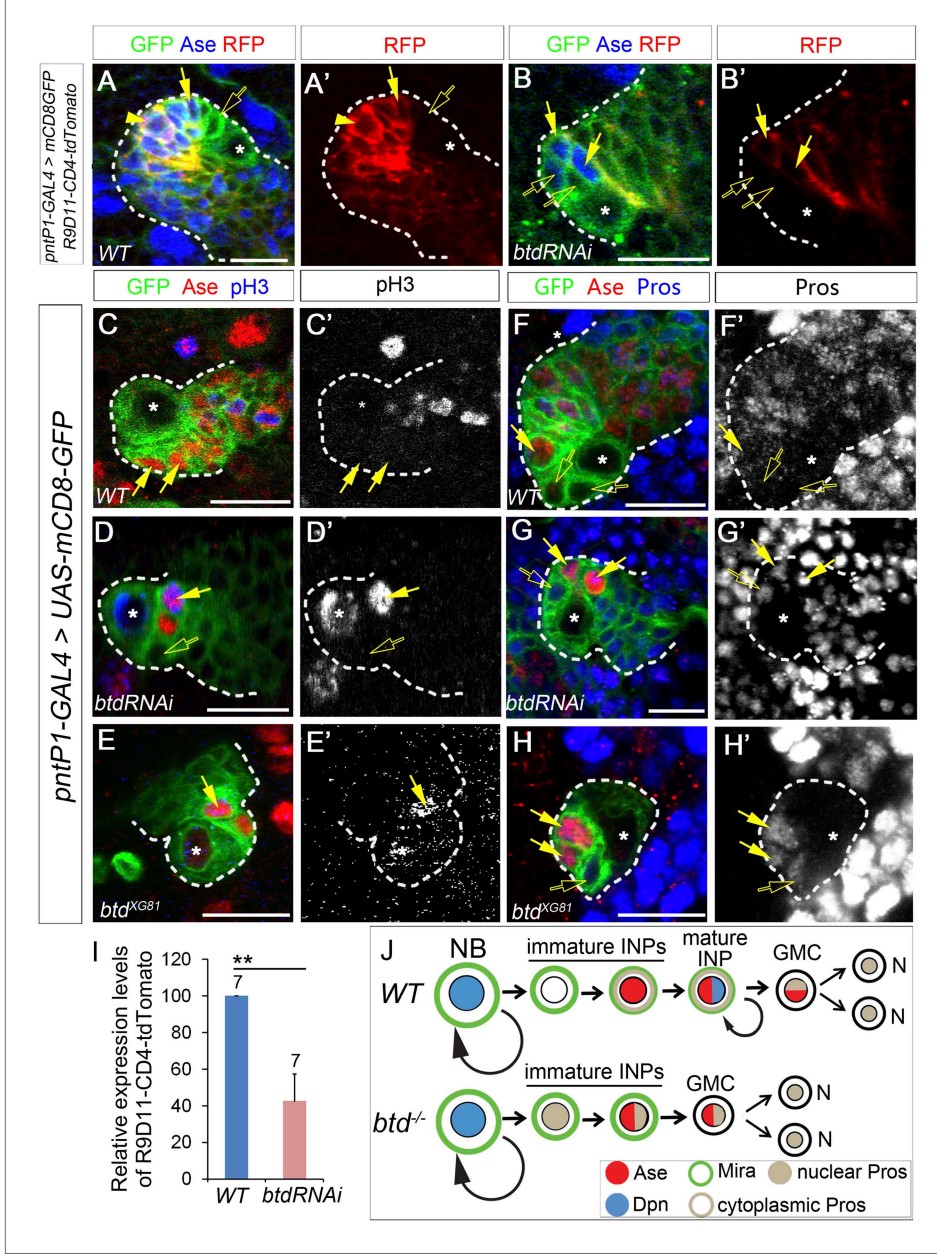

**Figure 2**. Loss of Btd results in ectopic nuclear Pros in immature INPs and premature differentiation of Ase⁺ immature INPs into GMCs. (**A–A'**) R9D11-CD4-tdTomato is expressed in Ase⁺ immature INPs (solid arrows) and mature INPs (arrowheads) but not in Ase⁻ immature INPs (open arrows) in a wild-type type II NB lineage. (**B–B'**) R9D11-CD4-tdTomato remains expressed in Ase⁺ daughter cells (solide arrows) next to the Ase⁻ immature INPs (open arrows) when mature INPs are eliminated by Btd RNAi knockdown. (**C–C'**) pH3 is not detected in Ase⁺ immature INPs (solid arrows) in a wild-type type II NB lineage. (**D–E'**) pH3 is expressed in Ase⁺ daughter cells (solid arrows) that are the furthest from the Ase⁻ immature INPs (open arrows) in a Btd RNAi knockdown type II NB lineage without mature INPs (**D–D'**) or *btd* mutant type II NB lineages (**E–E'**). (**F–F'**) Nuclear Pros is not expressed in Ase⁻ immature INPs (open arrows) in a wild-type type II NB lineage. (**G-H'**) Nuclear Pros is ectopically expressed in both Ase⁻ immature INPs (open arrows) and Ase⁺ cells (solid arrows) in Btd RNAi knockdown (**G–G'**) or btd mutant (**H–H'**) type II NB lineages. (**I**) Quantifications of relative overall expression levels of R9D11-CD4-tdTomato in wild-type (**A–A'**) and Btd RNAi knockdown (**B–B'**) type II NB lineages. (**J**) A diagram of neurogenesis patterns in type II NB lineages in the presence or absence of Btd.

The following figure supplement is available for figure 2:

**Figure supplement 1**. Mira expression in Btd RNAi knockdown type II NB clone.

Btd, we expected that some of the Ase[+] daughter cells express nuclear Pros. Interestingly, immunostaining of Pros showed that nuclear Pros was expressed not only in all Ase[+] daughter cells but also in Ase[−] immature INPs generated in Btd RNAi knockdown or *btd* mutant type II NB lineages (*Figure 2G–H'*). Given that Pros promotes cell cycle exit and GMC differentiation and that forced expression of Pros is sufficient to eliminate INPs in type II NB lineages (*Li and Vaessin, 2000*; *Choksi et al., 2006*; *Bayraktar et al., 2010*), the ectopic expression of nuclear Pros in immature INPs resulting from the loss of Btd very likely promotes the premature differentiation of Ase[+] immature INPs into GMCs and cell cycle exit, leading to the loss of mature INPs (*Figure 2J*). These results also reveal that it is Btd that is responsible for the suppression of Pros in immature INPs.

## Reducing Pros expression rescues the elimination of mature INPs resulting from the loss of Btd

To determine if the ectopic expression of nuclear Pros in immature INPs is indeed responsible for the elimination of mature INPs in the absence of Btd, we next examined if reducing Pros expression was able to rescue the elimination of INPs in Btd RNAi knockdown or *btd* mutant type II NB lineages. To reduce Pros expression, we either removed one wild-type copy of *pros* or knocked down Pros by RNAi in type II NB lineages. Remarkably, the elimination of mature INPs resulting from the Btd RNAi knockdown was nearly fully rescued even just by removing one wild-type copy of *pros* (*Figure 3A–D',I–J*). Unlike Btd RNAi knockdown in wild-type background, which resulted in a completely elimination of mature INPs in about 50% of type II NB lineages (*Figure 1B–B',G–H*, *Figure 3B–B',I–J*), knockdown of Btd in *pros[17]* or *pros[10419]* heterozygous mutant animals no longer led to an obvious loss of mature INPs (*Figure 3D–D',I–J*, and data not shown), although type II NB lineages develop normally in *pros[17]* or *pros[10419]* heterozygous mutant animals (*Figure 3C–C',I–J*, and data not shown). Similarly, the loss of INPs in *btd* mutant clones was also largely rescued when *btd* mutant type II NB clones were generated in *pros[17]* heterozygous mutant background (*Figure 3—figure supplement 1*).

Consistent with the rescue in *pros* heterozygous mutant animals, Pros RNAi knockdown also rescued the loss of INPs resulting from the loss of Btd (*Figure 3E–H', K–L*, *Figure 3—figure supplement 1*). Pros RNAi knockdown led to overproliferation of mature INPs as observed in *pros* mutant type II NB clones (*Figure 3G–G',K*, *Figure 3—figure supplement 1*) (*Bowman et al., 2008*). When Pros was knocked down in *btd* mutant type II NB clones, mature INPs were rescued in all *btd* mutant clones (*Figure 3H–H',L*). In about 70% of *btd* mutant type II NB clones, Pros RNAi knockdown led to a similar mature INP overproliferation as in wild-type clones. In other 30% of *btd* mutant clones, Pros RNAi knockdown partially or fully rescued mature INPs without causing the overproliferation of mature INPs. Similarly, Pros RNAi knockdown also rescued the loss of mature INPs resulting from Btd RNAi knockdown (*Figure 3—figure supplement 1*). Results from these rescue experiments demonstrate that the ectopic nuclear Pros in immature INPs is indeed responsible for the loss of mature INPs.

Interestingly, removing one wild-type copy of *pros* or knocking down Pros not only rescued the loss of mature INPs but also suppressed the ectopic Ase expression in all *btd* mutant type II NBs (*Figure 3H*, *Figure 3—figure supplement 1*), suggesting that the ectopic Ase expression in *btd* mutant type II NBs also results from the ectopic expression of nuclear Pros in immature INPs.

## Btd likely functions only in newly born immature INPs

Our results showed that Btd is required to suppress Pros in Ase[−] immature INPs. We next asked if Btd is also required to partially suppress Pros at later stages of INP development. In normal type II NB lineages, Pros is absent in Ase[−] immature INPs but is expressed at low levels in the cytoplasm of Ase[+] immature INPs and mature INPs (*Bello et al., 2008*; *Boone and Doe, 2008*; *Bowman et al., 2008*). Maintaining the expression of Pros at low levels is essential for the self-renewal of INPs (*Bayraktar et al., 2010*). However, the complete elimination of mature INPs makes it difficult to assess the role of Btd in mature INPs. Therefore, we used *erm-GAL4 (III)* and *erm-GAL4 (II)* to knock down Btd. Both *erm-GAL4 (III)* and *erm-GAL4 (II)* are expressed in Ase[+] immature INPs and mature INPs, whereas *erm-GAL4 (II)* is also expressed in Ase[−] immature INPs except for the newly born Ase[−] immature INPs (*Xiao et al., 2012*). However, knockdown of Btd using either *erm-GAL4 (III)* (*Figure 4A–B'',E*) or *erm-GAL4 (II)* (*Figure 4C–D'',F*) did not result in any obvious loss of mature INPs in type II NB lineages. In line with these RNAi knockdown results, we were able to recover multicellular *btd* mutant INP clones that were comparable to wild-type INP clones (*Figure 4—figure supplement 1*) while we generated *btd* mutant type II NB clones, indicating that *btd* mutant INPs were still able to divide multiple rounds like

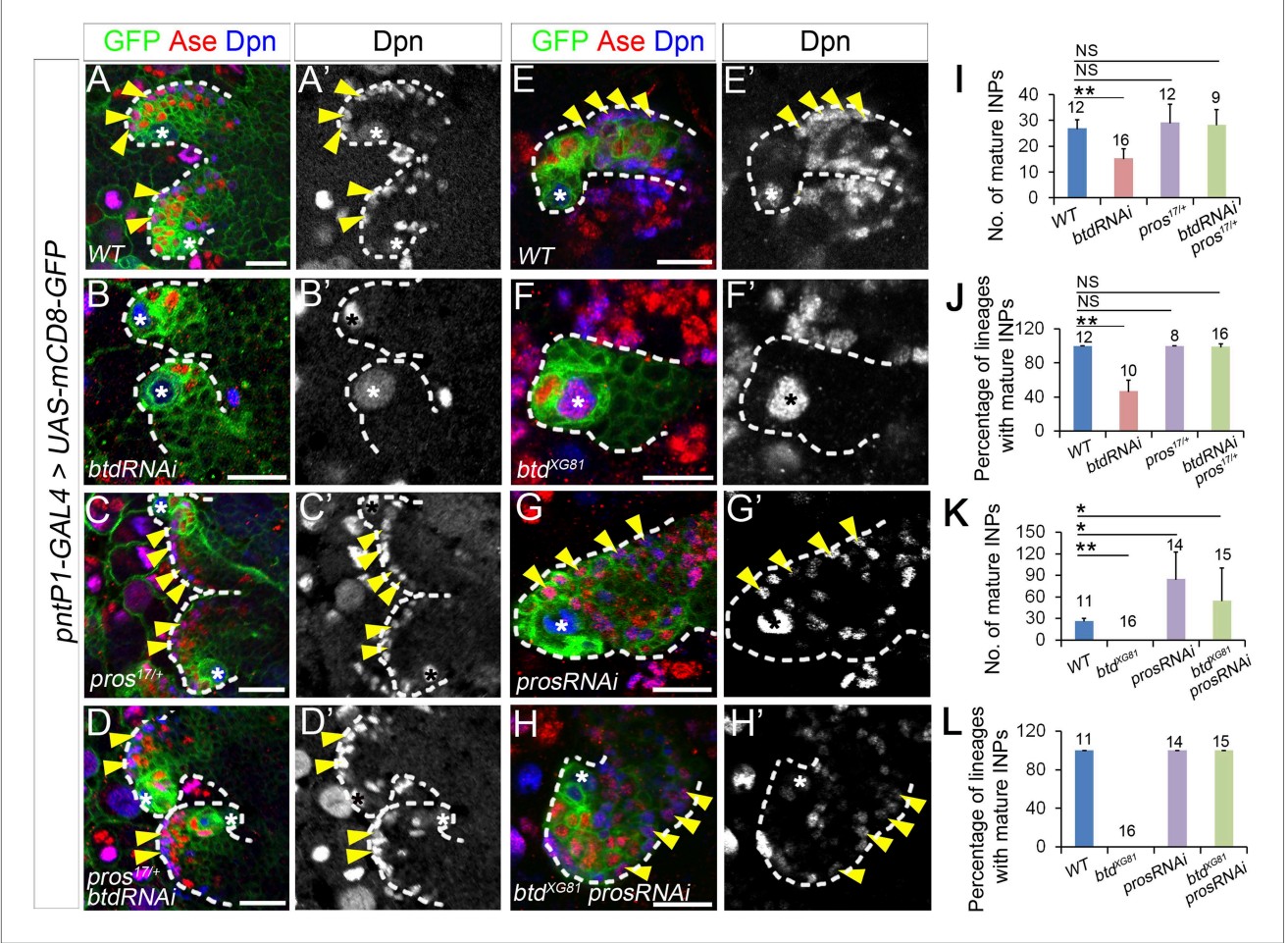

**Figure 3**. Reducing Pros rescues the elimination of INPs resulting from the loss of Btd. (**A–D'**) The loss of INPs resulting from Btd RNAi knockdown is rescued in *pros$^{17}$* heterozygous mutant background. Only two lineages are shown in each brain. (**A–A'**) Wild-type type II NB lineages have multiple mature INPs. (**B–B'**) Btd RNAi knockdown causes a loss of mature INPs. (**C–C'**) Type II NB lineages in *pros$^{17}$* heterozygous mutant larvae produce a similar number of mature INPs as in wild-type larvae. (**D–D'**) Btd RNAi knockdown no long leads to the loss of mature INPs in *pros$^{17}$* heterozygous mutant type II NB lineages. (**E–H'**) Pros RNAi knockdown rescues the loss of INPs in btd mutant type II NB clones. (**E–E'**) A wild-type type II NB clone has multiple mature INPs. (**F–F'**) A btd mutant type II NB clone contains no mature INPs. (**G–G'**) Pros RNAi knockdown causes overproliferation of mature INPs in a type II NB clone. (**H–H'**) Pros RNAi knockdown rescues the loss of mature INPs in a *btd* mutant type II clone. Arrowheads point to mature INPs in all images. (**I–L**) Quantifications of the number of mature INPs (**I–K**) and the percentage of lineages with mature INPs (**J–L**) for the rescue of Btd RNAi knockdown phenotypes in *pros$^{17}$/+* larvae (**I–J**) or the rescue of *btd* mutant phenotypes by Pros RNAi knockdown (**K–L**). The samples sizes on top of each bar represent the number of type II NB lineages (**I, K, L**) or the number of brain lobes (**J**). The mean and stdev in (**L**) are calculated by bootstrapping. **p < 0.01, *p < 0.05 (Student *t* test). NS: not significant.

The following figure supplement is available for figure 3:

**Figure supplement 1**. Reducing Pros rescues the elimination of INPs resulting from the loss of Btd.

wild-type INPs and did not prematurely differentiate into GMCs. These data suggest that Btd likely suppresses Pros expression only in newly born Ase⁻ immature INPs but not in immature INPs at later developmental stages or mature INPs.

## Reduction of PntP1 expression is responsible for the ectopic Ase expression but not the loss of mature INPs in btd mutant type II NB lineages

Our *btd* mutant MARCM analyses showed that in addition to the loss of INPs, Ase was ectopically expressed in about 40% of *btd* mutant type II NBs. We showed previously that PntP1 is expressed in

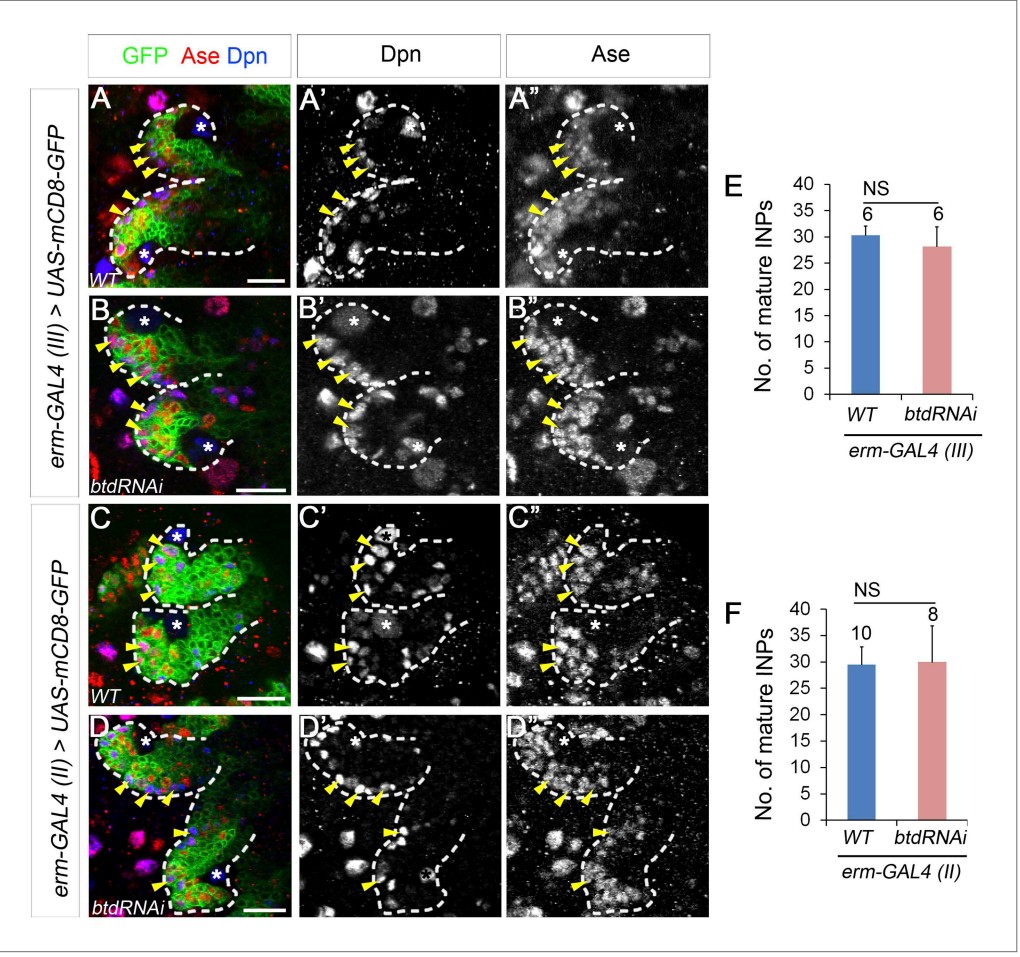

**Figure 4**. Knockdown of Btd in immature or mature INPs by *erm-GAL4* lines does not lead to the loss of mature INPs. (**A–A''**, **C–C''**) Wild-type type II NB lineages are labeled with mCD8-GFP driven by *erm-GAL4* (III) (**A–A''**) or *erm-GAL4* (II) (**C–C''**). (**B–B''**, **D–D''**) Knockdown of Btd in Ase⁺ immature INPs and mature INPs by *erm-GAL4* (III) (**B–B''**) or in Ase⁻ immature INPs as well as Ase⁺ immature INP and mature INPs by *erm-GAL4* (II) (**D–D''**) does not cause a reduction of the number of mature INPs. Only two lineages are shown in each brain. (**E–F**) Quantifications of the number of mature INPs in type II NB lineages in which Btd is knocked down by *erm-GAL4* (III) (**E**) or *erm-GAL4* (II) (**F**). NS: not significant.

The following figure supplement is available for figure 4:

**Figure supplement 1**. Btd likely does not function in mature INPs.

type II NBs as well as Ase⁻ and Ase⁺ immature INPs (***Figure 5A–A''***) (***Zhu et al., 2011***). Inhibiting PntP1 activity results in ectopic Ase expression in type II NBs and elimination of INPs (***Zhu et al., 2011***). Therefore, we wondered if PntP1 expression was reduced or even lost in *btd* mutant type II NBs. Immunostaining of PntP1 showed that PntP1 was expressed at reduced levels in most *btd* mutant type II NB lineages without the ectopic Ase expression (***Figure 5B–B'***). The reduction is about 10% in the NBs and 50% in the immature INPs (***Figure 5E–F***). In those *btd* mutant clones, PntP1 was also detected in the Ase⁺ daughter cells next to the Ase⁻ immature INPs (***Figure 5B–B'***), providing additional evidence to support that Ase⁻ immature INPs still differentiate into Ase⁺ immature INPs in the absence of Btd. However, in *btd* mutant type II NB lineages with the ectopic Ase expression in the NB, PntP1 was largely abolished in both the NBs and their progeny (***Figure 5C–C',E–F***). The correlation of the ectopic Ase expression in the NB and the severe reduction or loss of PntP1 suggesting that the ectopic Ase expression in *btd* mutant type II NBs could result from the severe reduction or loss of PntP1 expression.

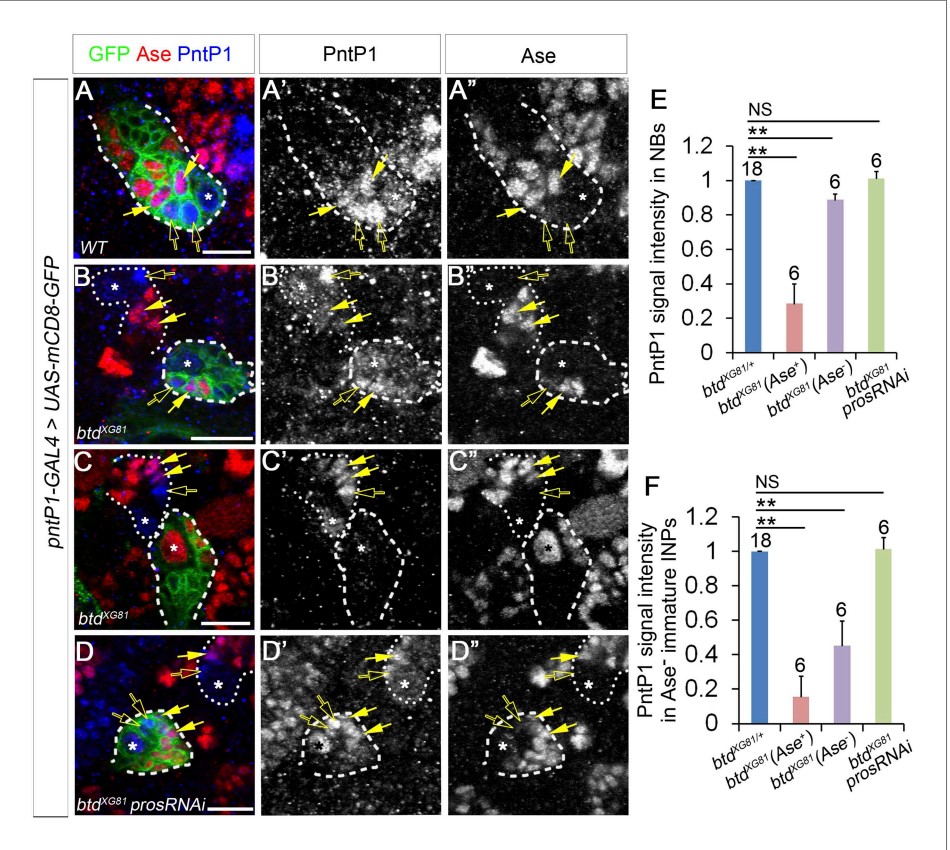

**Figure 5**. PntP1 expression is reduced in btd mutant type II NB clones. (**A–A''**) PntP1 is expressed in the NB (*), Ase⁻ immature INPs (open arrows), as well as Ase⁺ immature INPs (solid arrows) in a wild-type type II NB clone. (**B–B''**) PntP1 expression is much lower in a *btd* mutant type II NB clone without the ectopic Ase expression in the NB than that in a neighboring *btd* heterozygous type II NB lineage. The reduction is particularly obvious in the Ase⁻ immature INPs (open arrows). Note that PntP1 remains expressed in Ase⁺ daughter cells (arrows) next to the Ase⁻ immature INPs. (**C–C''**) PntP1 expression is largely abolished in a *btd* mutant type II NB clone with the ectopic Ase expression in the NB. In a neighboring *btd* heterozygous type II NB lineages, PntP1 is still detected in the NBs (*), Ase⁻ immature INPs (open arrows) and Ase⁺ immature INPs (arrows). (**D–D''**) Knocking down Pros restores the expression of PntP1 in the NB (*), Ase⁻ immature INPs (open arrows), and Ase⁺ immature INPs (arrows) in a *btd* mutant clone to levels comparable to those in a neighboring btd heterozygous type II NB lineage. Wild-type (**A–A''**) or *btd* mutant type II NB clones (**B–D''**) are outlined by dashed lines and neighboring *btd* heterozygous type II NB lineages (**B–D''**) are marked with dotted lines. (**E–F**) Quantifications of PntP1 expression levels in type II NBs (**E**) and Ase⁻ immature INPs (**F**) in *btd* mutant type II NB clones relative to neighboring type II NB lineages in the same brains. **p < 0.01; NS: not significant.

The following figure supplement is available for figure 5:

**Figure supplement 1**. The reduction of PntP1 is unlikely responsible for the loss of INPs in btd mutant type II NB clones.

To determine if the reduction/loss of PntP1 is responsible for the ectopic Ase expression and/or the loss of INPs in *btd* mutant type II NB lineages, we examined if restoring PntP1 expression was sufficient to suppress ectopic Ase expression and/or rescue the loss of INPs resulting from the loss of Btd by expressing *UAS-pntP1* in *btd* mutant type II NB clones. Our results showed that expressing *UAS-pntP1* resulted in higher expression of PntP1 in *btd* mutant type II NB clones than that in neighboring *btd* heterozygous mutant type II NB lineages and suppressed the ectopic Ase expression in all *btd* mutant type II NBs (n = 12) (***Figure 5—figure supplement 1***). However, unlike reducing Pros expression, which rescued mature INPs in nearly all *btd* mutant type II NB clones (***Figure 3***, ***Figure 3—figure supplement 1***), expressing *UAS-pntP1* failed to rescue mature INPs or suppress the ectopic nuclear

Pros in Ase⁻ immature INPs or Ase⁺ daughter cells in the majority of *btd* mutant clones (***Figure 5—figure supplement 1***). Only in 3 out of total 10 *btd* mutant clones expressing *UAS-PntP1*, we observed that mature INPs were partially rescued to 9.3 ± 3.2 per lineages (***Figure 5—figure supplement 1***), which is still much fewer than the number of mature INPs (20–30 per lineages) in normal type II NB lineages. Consistent with the inability of *UAS-pntP1* to fully rescue the loss of mature INPs, we found occasionally that *btd* mutant clones that did not show an obvious reduction of PntP1 in either the NBs or early immature INPs still failed to generate any mature INPs (***Figure 5—figure supplement 1***). Therefore, these results demonstrate that the severe reduction/loss of PntP1 accounts for the ectopic Ase expression in *btd* mutant type II NBs but is not the primary reason for the loss of mature INPs.

Given that reducing the expression of Pros suppressed the ectopic Ase expression in *btd* mutant type II NBs, we then asked if reducing Pros expression could also rescue the reduction/loss of PntP1 in *btd* mutant type II NB clones. Indeed, consistent with the suppression of the ectopic Ase expression by Pros RNAi knockdown, PntP1 expression in both the NBs and immature INPs returned to normal levels when the loss of INPs was rescued by Pros RNAi knockdown in *btd* mutant type II NB clones (***Figure 5D–F***). These results suggest that the reduction/loss of PntP1 in *btd* mutant type II NB clones is due to the ectopic Pros expression in immature INPs. However, given that ectopic nuclear Pros is only observed in immature INPs but not in the NBs in *btd* mutant type II NB clones, the reduction/loss of PntP1 and the subsequent ectopic Ase expression in the NB is most likely a secondary effect of the ectopic nuclear Pros expression in immature INPs.

## Btd is expressed in type II NB lineages and a subset of type I NB lineages

Our Btd loss of function analyses demonstrated that Btd is critical for the generation of INPs in type II NBs lineages. We next examined if Btd is only expressed type II NB lineages. Since Btd antibodies are not available and our in situ hybridization signals of btd mRNAs in the central brain were barely detectable (data now shown), we used the *btd-GAL4* as a reporter for *btd* expression. *btd-GAL4* is a *GAL4* enhancer trap line, in which the *GAL4* transgene is inserted at 753bp upstream of the transcription start site of *btd* (***Estella et al., 2003***). *btd-GAL4* shows similar expression patterns as endogenous Btd in ventral imaginal discs (***Estella et al., 2003***). We found that mCD8-GFP driven by the *btd-GAL4* is expressed in all type II NB lineages but not type I NB lineages on the dorsal side of larval brains (***Figure 6A–A′***). In type II NB lineages, the expression of mCD8-GFP driven by *btd-GAL4* is detected in the NB but becomes much stronger in immature INPs next to the NBs (***Figure 6A***). The expression of mCD8-GFP then becomes progressively weak in cells away from the NBs and is barely detectable in some mature INPs distal from the NB (***Figure 6A–A′***). The expression pattern of *btd-GAL4* in type II NB lineages is similar to that of *pntP1-GAL4* (e.g. ***Figure 1A–A′***) and is consistent with our results that Btd mainly functions in immature INPs.

In addition to type II NB lineages, mCD8-GFP driven by *btd-GAL4* is also expressed in two type I NB lineages on the ventral side of larval brains as well as about 31 type I NB lineages in the ventral nerve cord (VNC) (***Figure 6B–C′***). In those type I NB lineages, mCD8-GFP driven by *btd-GAL4* is expressed in the NBs, GMCs, and newly born neurons. The expression pattern of *btd-GAL4* in single type I NB lineages is similar to those of other NB GAL4 lines such as *insc-GAL4* (e.g. ***Figure 7D***), suggesting that Btd likely functions in the NB in these type I NB lineages.

In order to determine if the *btd-GAL4* reflects the endogenous Btd expression pattern in the central brain, we tried to rescue the *btd* loss-of-function phenotypes by expressing *UAS-mSp8* or *UAS-btd* driven by *btd-GAL4*. However, the *GAL4* insertion in the *btd-GAL4* line does not affect the type II NB lineage development although it causes a lethal mutation of *btd* (***Figure 6—figure supplement 1***). Therefore, we tried to rescue Btd RNAi knockdown phenotypes by the expression of *UAS-mSp8* or *UAS-btd* driven by *btd-GAL4* instead. The expression of *UAS-btd RNAi* driven by *btd-GAL4* completely eliminated mature INPs in nearly all type II NB lineages (***Figure 6—figure supplement 2***), which was much stronger than the phenotype of Btd RNAi knockdown driven by *pntP1-GAL4*. As expected, the expression of *UAS-mSp8* driven by *btd-GAL4* fully rescued the loss of INPs resulting from Btd RNAi knockdown in all type II NB lineages (***Figure 6—figure supplement 2***). Similarly, the expression of *UAS-btd* driven by *btd-GAL4* partially rescued the loss of INPs in about 67% of lineages (***Figure 6—figure supplement 2***). The incomplete rescue by *UAS-btd* is likely because *UAS-btd* contains the sequence targeted by *UAS-btd RNAi*. The rescue of Btd RNAi knockdown phenotypes by the expression of *UAS-mSp8* and *UAS-btd* driven by *btd-GAL4* together with the strong loss of INP

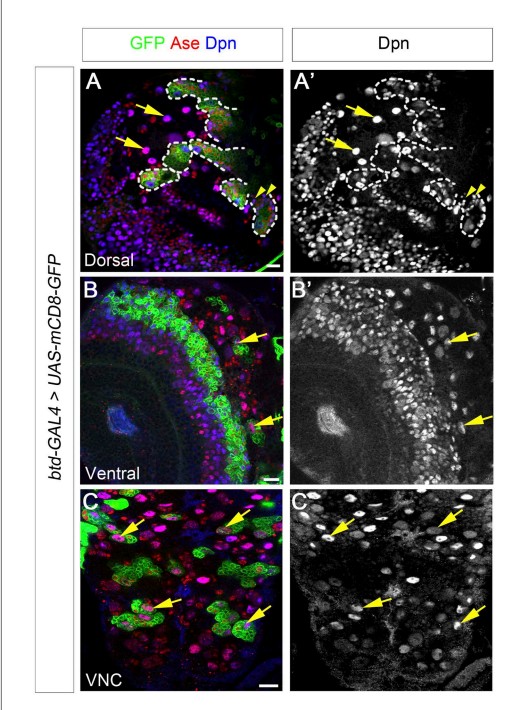

**Figure 6**. Btd is expressed in type II NB lineages and a subset of type I NB lineages. (**A**–**A**') mCD8-GFP driven by *btd-Gal4* is expressed in all type II NB lineages (outlined by dashed lines) but not type I NB lineages (e.g. arrows) on the dorsal side of a 3rd instar larval brain. The expression of mCD8-GFP becomes progressively weak in cell away from the NB. Some mature INPs (e.g. arrowheads) distant from the NB have no obvious expression of mCD8-GFP. Only seven out of total eight type II NB lineages are shown in this particular focal plane. (**B**–**B**') Two type I NB lineages are labeled by mCD8-GFP driven by *btd-GAL4* on the ventral side of a 3rd instar larval brain. (**C**–**C**') mCD8-GFP driven by *btd-Gal4* labels a subset of type I NB lineages (e.g. arrows) in the ventral nerve cord (VNC).

The following figure supplements are available for figure 6:

**Figure supplement 1**. The GAL4 insertion in the btd-GAL4 line does not affect type II NB lineage development.

**Figure supplement 2**. The loss of INP phenotype resulting from Btd RNAi knockdown is rescued by the expression of mouse Sp8 (mSp8) or Drosophila Btd.

phenotypes in all *btd* mutant type II NB clones strongly argue that *btd-GAL4* expression is likely consistent with the endogenous Btd expression pattern in the central brain.

## PntP1 and Btd function cooperatively to promote the generation of INPs

Our previous studies showed that ectopic expression of PntP1 could induce the generation of INP-like cells in more type I NB lineages in VNCs than in larval brains (*Figure 7H–H',J*, *Figure 7—figure supplement 1*) (*Zhu et al., 2011*). Since Btd is expressed in much more type I NB lineages in VNCs than in larval brains and Btd is required to prevent the premature differentiation of INPs, we wondered if ectopic PntP1-induced generation of INP-like cells requires Btd activity and occurs mostly in Btd-positive type I NB lineages. To test this idea, we examined if co-expression of PntP1 and Btd in type I NB lineages was sufficient to induce the generation of INPs and if the ectopic PntP1-induced generation of INP-like cells would be impaired in the absence of Btd.

To coexpress PntP1 and Btd in type I NB lineages, we used either *btd-GAL4* to drive the expression of *UAS-pntP1* in Btd-positive type I NB lineages or *insc-GAL4* to drive the expression of *UAS-pntP1* and *UAS-btd* simultaneously in all type I NB lineages. INP-like cells were identified by their expression of Ase and Dpn as well as INP-specific marker R9D11-CD4-*tdTomato*. Since the *GAL4* insertion in the *btd-GAL4* line causes a lethal mutation in *btd* (*Estella et al., 2003*), we examined the phenotype of the expression of *UAS-pntP1* driven by *btd-GAL4* only in *btd-GAL4* heterozygous female larvae. As shown in *Figure 6A–A'*, type II NB lineages in *btd-GAL4* heterozygous mutant larvae are indistinguishable from those in wild-type animals (e.g. *Figure 1A–A'*). Furthermore, as mentioned above, *btd-GAL4* homozygous mutant type II NB clones develop normally (*Figure 6—figure supplement 1*). Therefore, the generation of INPs is not affected in the *btd-GAL4* line. Interestingly, ectopic expression of PntP1 using *btd-GAL4* as a driver induced the generation of INP-like cells in about 90% Btd-positive type I NB lineages in both larval brains (*Figure 7A–C',J*) and VNCs (*Figure 7—figure supplement 1*). Consistently, the co-expression of

*UAS-pntP1* and *UAS-btd* driven by *insc-GAL4* induced INP-like cells in about 95% of type I NB lineages in both larval brains and VNCs (*Figure 7F–F',I–I',J*, *Figure 7—figure supplement 1*). In contrast, the expression of *UAS-pntP1* alone driven by *insc-GAL4* only induced INP-like cells in about 10% and 46% of type I NB lineages in larval brains (*Figure 7H–H',J*) and VNCs (*Figure 7—figure supplement 1*), respectively, although ectopic PntP1 expression suppressed Ase in nearly all type I NBs (*Figure 7H*, *Figure 7—figure supplement 1*). The expression of *UAS-btd* alone neither suppressed Ase nor

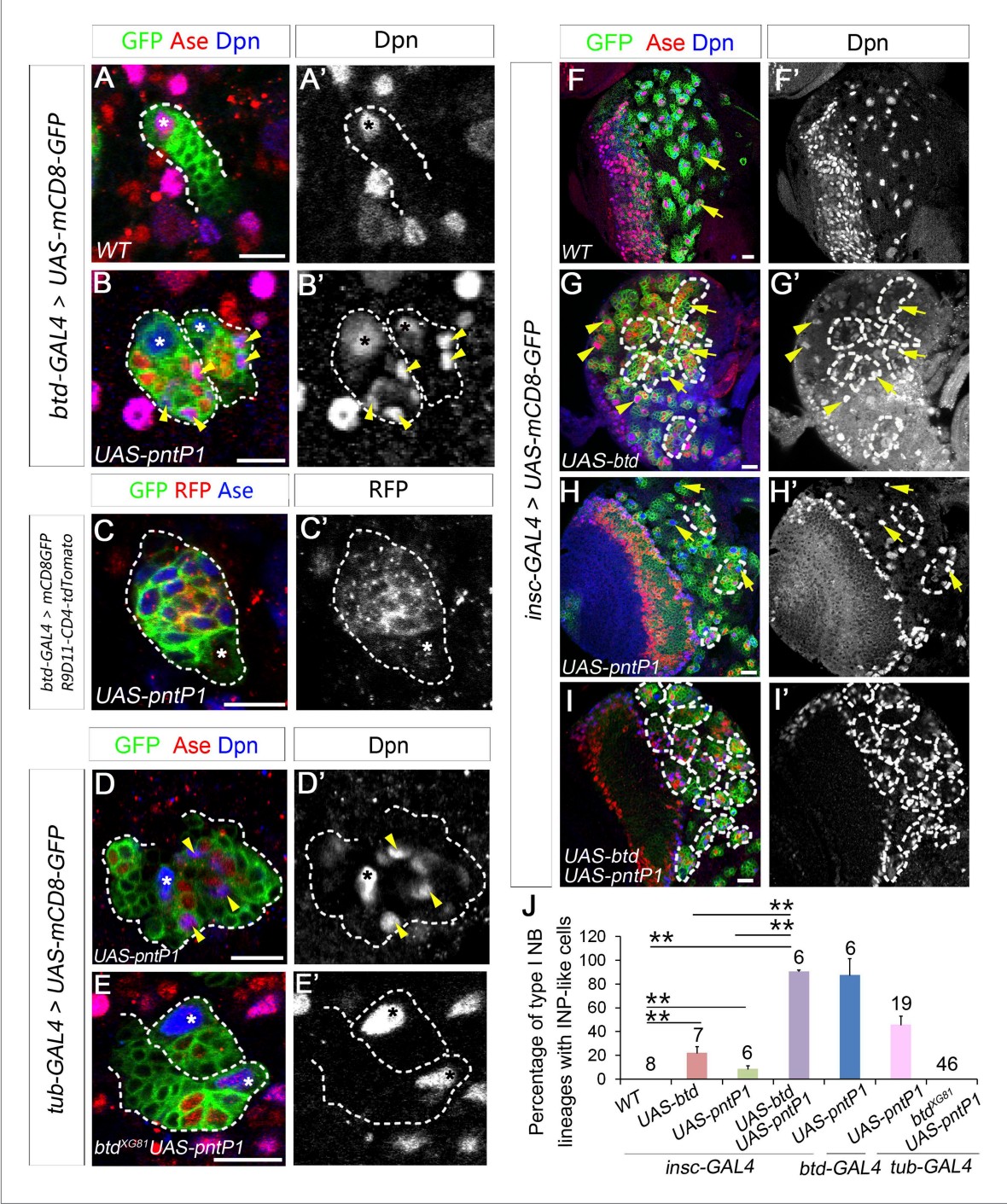

**Figure 7**. Btd and PntP1 function cooperatively to promote the generation of INPs. (**A–C'**) Ectopic expression of PntP1 consistently promotes the generation of INP-like cells in Btd-positive type I NB lineages in larval brains. (**A–A'**) A wild-type type I NB lineage labeled by *btd-GAL4* has no INP-like cells. (**B–C'**) The ectopic expression of PntP1 driven by *btd-GAL4* suppresses Ase in the NB (*) and induces the generation of Ase⁺ Dpn⁺ INP-like cells (arrowheads) (**B–B'**), which also express INP-specific marker R9D11-CD4-tdTomato (**C–C'**). (**D–E'**) The expression of PntP1 driven by *tub-GAL4* suppresses Ase in both wild-type (**D**) and *btd* mutant (**E**) type I NBs (*) but only induced the generation of INP-like cells in the wild-type type I NB clone (**D–D'**) but not in the *btd* mutant clone (**E–E'**). (**F–I'**) *insc-GAL4* drives the expression of *UAS-mCD8-GFP* alone (**F–F'**) or together with *UAS-btd* (**G–G'**), *UAS-pntP1* (**H–H'**), or *UAS-btd* plus *UAS-pntP1* (**I–I'**) in type I NB lineages. Images are from the ventral side of larval brains, where only type I NB lineages are observed in wild-type animals (**F–F'**). The expression of *UAS-btd* (**G–G'**) or *UAS-pntP1* (**H–H'**) alone promotes the generation of INP-like cells only in small subset of type I NB lineages (dashed circles). The expression of Btd only suppresses/reduces Ase expression in type I NBs (arrows) that produce INP-like cells (**G–G'**) but not in other type I NBs (arrowheads), where PntP1 suppresses Ase in nearly all type I NBs (e.g. arrows) regardless of the generation of

*Figure 7. Continued on next page*

*Figure 7. Continued*

INP-like cells (**H**–**H'**). Co-expression of Btd and PntP1 promotes the generation of INP-like cells nearly in all type I NB lineages (dashed circles) (**I**–**I'**). (**J**–**J'**) Quantifications of the percentage of type I NB lineages that produce INP-like cells in larvae with indicated genotypes. The number on top of each bar represents the number of brain lobes except for numbers for the expression of *UAS-pntP1* driven by *tub-GAL4*, which are the number of clones. The mean and stdev for the percentage of wild-type or *btd* mutant clones expressing *UAS-pntP1* driven by *tub-GAL4* are calculated by bootstrapping. **p < 0.01.

The following figure supplements are available for figure 7:

**Figure supplement 1**. Btd and PntP1 function cooperatively to induce the generation of INP-like cells in type I NB lineages in the VNC.

**Figure supplement 2**. Overexpression of Btd promotes the generation of INP-like cells.

induced the generation of INP-like cells in VNCs (*Figure 7—figure supplement 1*), whereas in larval brains, the expression of *UAS-btd* driven by *insc-GAL4* suppressed/reduced the expression of Ase in the NB and promoted the generation of INP-like cells in about 20% of type I NB lineages (*Figure 7G–G',J*). These ectopic INP-like cells induced by the expression of *UAS-btd* also expressed INP-specific marker R9D11-CD4-*tdTomato* (*Figure 7—figure supplement 2*). These results indicate that the generation of INP-like cells induced by the ectopic PntP1 expression requires Btd activity, whereas the expression of either PntP1 or Btd alone has limited ability to induce the generation of INP-like cells in type I NB lineages.

To further confirm if Btd is required for ectopic PntP1 to induce the generation of INP-like cells, we then examined if the generation of INP-like cells induced by ectopic PntP1 expression would be impaired in the absence of Btd. To this end, we expressed *UAS-pntP1* in wild-type or *btd* mutant type I NB clones in VNCs in that there are more Btd-positive type I NB lineages and the expression of *UAS-pntP1* can induce INP-like cells in much more type I NB lineages in VNCs than in larval brains. Consistent with the induction of INP-like cells in nearly all type I NB lineages when PntP1 and Btd were coexpressed, the efficiency of PntP1 to induce the generation of INP-like cell was drastically reduced in the absence of Btd. Our results showed that the expression of *UAS-pntP1* driven by *tub-GAL4* could have induced the generation of INP-like cells in about 50% of wild-type type I NB clones but not in *btd* mutant type I NB clones, although the expression of PntP1 equally suppressed the expression of Ase in both wild-type and *btd* mutant type I NBs (*Figure 7D–E',J*). These results provide additional evidence to support that only in the presence of Btd could PntP1 induce the generation of INP-like cells in type I NB lineages.

Taken together, these results suggest that the generation of INPs requires the cooperative action of PntP1 and Btd. Thus this study together with our previous work (*Zhu et al., 2011*) identified two key factors, PntP1 and Btd, a combination of which is sufficient to specify type II NB lineages and promote INP generation.

## Discussion

In this study, we show that the Sp family transcription factor Btd is required to prevent the premature differentiation of INPs by suppressing the expression of Pros in immature INPs. Furthermore, we provide evidence to demonstrate that the combination of Btd and PntP1 is sufficient to specify type II NB lineages and promote the generation of INPs. Thus, our work reveals a critical mechanism that regulates INP generation.

### Btd prevents premature differentiation of INPs into GMCs

The most striking phenotype resulting from the loss of Btd is the elimination of mature INPs. In addition, about 40% of *btd* mutant type II NB lineages ectopically express Ase in the NB and become type I-like NB lineages. However, although forced expression of Ase in type II NBs is sufficient to eliminate INPs in type II NB lineages (*Bowman et al., 2008*; *Zhu et al., 2012*), the loss of INPs is obviously not primarily due to the ectopic Ase expression or the transformation of type II NB lineages into type I-like NB lineage in that the loss of mature INPs occurs independently of the ectopic Ase expression in most *btd* mutant or Btd RNAi knockdown type II NB lineages. Instead, we demonstrate that the loss of mature INPs in the absence of Btd is due to the premature differentiation of Ase$^+$ immature INPs into GMCs. We show that in Btd RNAi knockdown or *btd* mutant type II NB lineages without the ectopic

Ase expression, Ase⁻ immature INPs differentiate into Ase⁺ immature INPs normally as indicated by the expression of R9D11-mCD8-GFP, Mira, as well as PntP1 in Ase⁺ daughter cells next to the Ase⁻ immature INPs. However, instead of differentiating into mature INPs, we argue that Ase⁺ immature INPs prematurely differentiate into GMCs based on the following two pieces of evidence. First, Ase⁺ daughter cells eventually undergo terminal divisions as indicated by the positive pH3 staining and the position of the pH3-positive cells. Second, unlike mature INPs, the dividing Ase⁺ daughter cells do not form basal Mira crescent at metaphase. The terminal division and the lack of Mira crescent during the division are two unique features that distinguish GMCs from INPs in addition to the expression of nuclear Pros (*Ikeshima-Kataoka et al., 1997*; *Matsuzaki et al., 1998*; *Schuldt et al., 1998*). Therefore, the elimination of mature INPs resulting from the loss of Btd is due to the premature differentiation of Ase⁺ immature INPs into GMCs.

Why does the loss of Btd lead to premature differentiation of INPs? Our results show that the loss of Btd results in a reduction or loss of PntP1 in type II NBs and immature INPs as well as ectopic expression of Pros in early immature INPs. Our previous studies show that PntP1 suppresses Ase in type II NBs and that inhibiting PntP1 activity leads to ectopic expression of Ase in type II NBs and elimination of INPs (*Zhu et al., 2011*). Given that the ectopic Ase expression in *btd* mutant type II NBs is closely associated with the severe reduction or complete loss of PntP1 and that expression of *UAS-pntP1* largely suppresses the ectopic Ase expression in *btd* mutant type II NBs, the severe reduction or loss of PntP1 most likely accounts for the ectopic Ase expression in *btd* mutant type II NBs. However, although the loss of PntP1 could lead to the loss of INPs, we provide several lines of evidence to demonstrate that the elimination of INPs in *btd* mutant or Btd RNAi knockdown type II NB lineages is primarily due to the ectopic activation of Pros in immature INPs rather than the reduction or loss of PntP1. First, ectopic nuclear Pros is consistently expressed in Ase⁻ immature INPs when mature INPs are eliminated. Second, the loss of mature INPs can be fully rescued by Pros RNAi knockdown or even just by removing one wild-type copy of *pros*. Third, Pros RNAi knockdown also rescues the reduction of PntP1 and suppresses the ectopic Ase expression in *btd* mutant type II NBs. In contrast, the expression of *UAS-pntP1* fails to rescue mature INPs in most *btd* mutant type II NB lineages although it largely suppresses the ectopic Ase expression in the NBs. Furthermore, the complete elimination of mature INPs is also observed occasionally in *btd* mutant type II NB lineages without the reduction of PntP1. Therefore, the elimination of mature INPs resulting from the loss of Btd is primarily due to the ectopic Pros expression, which likely promotes premature differentiation of INPs into GMCs and cell cycle exit. The severe reduction or loss of PntP1 is responsible for the ectopic Ase expression in *btd* mutant type II NBs and is more likely a secondary effect due to the ectopic Pros expression and/or the loss of INPs. INPs and/or other progeny may provide feedback signals to the NBs as has been demonstrated in other systems (*Yoon et al., 2008*; *Hsu et al., 2014*).

The ectopic expression of Pros in Ase⁻ immature INPs resulting from the loss of Btd suggests that Btd is critical for suppressing Pros expression in Ase⁻ immature INPs. Btd was known as a head gap gene. It has been suggested that gap factors act largely as transcriptional repressors (*Schroeder et al., 2004*). Btd could directly suppress Pros by binding to the *pros* promoter as a transcriptional repressor. Alternatively, Btd could suppress Pros indirectly by regulating the expression or antagonizing the activity of factor(s) that activate(s) *pros* expression. Our results show that ectopic/overly expression of Btd in type I NB lineages or mature INPs does not lead to overproliferation of type I NBs as observed in *pros* mutant type I NB lineages. Instead, ectopic expression of Btd promotes the generation of INP-like cells from type I NBs and transforms some type I NB lineages into type II-like NB lineages. Therefore, it is more likely that Btd suppresses Pros indirectly by regulating the expression or antagonizing the activity of *pros* activator(s). Previous studies have suggested that Ase, Daughterless, Numb, and Erm could activate *pros* expression (*Reddy and Rodrigues, 1999*; *Southall and Brand, 2009*; *Weng et al., 2010*; *Yasugi et al., 2014*). Since Ase and R9D11-Cd4-tdTomato, which are under the control of *erm* promoter, are not expressed in Ase⁻ immature INPs in the absence of Btd, it is unlikely that they are involved in the activation of *pros* in immature INPs. It would be interesting to investigate in the future if Numb or Daughterless could activate *pros* in immature INPs in the absence of Btd.

## Btd and PntP1 function cooperatively to promote the generation of INPs

In this study, we provided several lines of evidence to demonstrate that Btd and PntP1 function cooperatively to specify type II NB lineages and promote the generation of INPs. Results from this study as well as our previous study (*Zhu et al., 2011*) show that ectopic expression of *UAS-pntP1* or *UAS-btd*

alone can only promote the generation of INP-like cells in a subset of type I NB lineage, whereas ectopic expression of *UAS-pntP1* in Btd-positive type I NB lineages or co-expression of *UAS-btd* and *UAS-pntP1* can promote the generation of INP-like cells in nearly all type I NB lineages and transforms all these lineages into type II-like NB lineages. Consistently, the ability of PntP1 to promote the generation of INP-like cells in *btd* mutant type I NB lineages is largely impaired. These results suggest that the specification of type II NB lineages and the generation of INPs requires both PntP1 and Btd and that the combinatorial PntP1 and Btd is sufficient to promote the generation of INPs.

We propose that PntP1 and Btd function cooperatively but through different mechanisms to promote INP generation. PntP1 is responsible for the suppression of Ase in type II NBs. Meanwhile, PntP1 must be regulating the expression of other unknown target gene(s) that are/is essential for the generation of INPs, such as specification of immature INPs, because loss of Ase is not sufficient to promote the generation of INP-like cells in any type I NB lineages. Btd likely acts after PntP1 to mainly prevent premature differentiation of INPs into GMCs by indirectly suppressing *pros* in immature INPs. The role of Btd in suppressing Ase in type II NBs is minimal if there is any because unlike PntP1, which suppresses *ase* in nearly all type I NBs when it is ectopically expressed, overexpression of Btd only suppresses Ase in a small subset of type I NBs that produce INP-like cells in larval brains. Furthermore, Ase is expressed in Btd+ type I NBs, indicating that Btd does not suppress Ase in type I NBs when it is expressed at normal levels. Studies in mammals as well as in *Drosophila* suggest that the Btd/Sp8 could function downstream of Wnt signaling to regulate the expression of Fgf8 as well as Distal-less (Dll) and Headcase (Hdc) during the forebrain patterning as well as limb development (*Estella et al., 2003*; *Treichel et al., 2003*; *Kawakami et al., 2004*; *Sahara et al., 2007*; *Estella and Mann, 2010*). However, inhibiting Wnt signaling alone in type II NB lineages does not have any obvious phenotypes (*Komori et al., 2014*), indicating that Btd unlikely functions downstream of Wnt signaling in type II NB lineages. Whether Fgf8, Dll, or Hdc could function downstream of Btd to regulate INP generation remains to be investigated in the future.

In mammals, the Btd homolog Sp8 plays important roles in brain development. In the developing mouse forebrain, Sp8 is expressed in cortical progenitors in a mediolateral gradient across the ventricular zone as well as in the lateral ganglionic eminence (LGE) and medial ganglionic eminence (MGE) (*Sahara et al., 2007*; *Zembrzycki et al., 2007*). In developing human brains, Sp8 is abundantly expressed in the ventricular zone and the outer sub-ventricular zone where RGs and oRGs reside (*Ma et al., 2013*). In addition to its roles in interneuron development and the patterning of developing mammalian brains and spinal cords (*Griesel et al., 2006*; *Waclaw et al., 2006*; *Sahara et al., 2007*; *Li et al., 2011*), it was also shown that the loss of Sp8 led to the reduction of the progenitor pool (*Zembrzycki et al., 2007*). Our results show that mammalian Sp8 can rescue the loss of mature INPs resulting from the loss of Btd in *Drosophila*, suggesting that Btd/Sp8 could have conserved functions across different species. It would be interesting to investigate if Sp8 has similar roles in promoting the generation of transient amplifying INPs, such as oRGs, in developing mammalian brains.

## Materials and methods

### Fly stocks

For *btd* loss-of-function analyses, *yw btd^XA FRT19A/FM7c* and *yw btd^XG81 FRT19A/FM7c* (*Wimmer et al., 1993*; *Estella and Mann, 2010*) were used for generating *btd* mutant clones and *UAS-btd RNAi* (#29453; Bloomington *Drosophila* stock Center, Bloomington, Indiana) for Btd RNAi knockdown. Type II NB lineage-specific *pntP1-GAL4* (also named as *GAL4^14−94*) (*Zhu et al., 2011*) and *erm-GAL4* (II) or (III) (*Pfeiffer et al., 2008*; *Xiao et al., 2012*) were used to drive the expression of *UAS-transgenes* in type II NB lineages or in immature as well as mature INPs, respectively. *insc-Gal4* (*Gal4^1407* inserted in *inscuteable* promoter) (*Luo et al., 1994*) was used to drive *UAS-transgenes* in all NB lineages. *UAS-mCD8-GFP* driven by *Btd-GAL4* was utilized as reporter for *btd* expression. The R9D11-CD4-tdTomato transgenic line (*Han et al., 2011*) was used for labeling Ase+ immature INPs and mature INPs. Other fly stocks include: *hs-Flpase tub-GAL80 FRT19A; UAS-mCD8-GFP; pntP1-GAL4* for generating type II NB clones; *pros^17/TM6 Tb, pros^10419/Tm3 Sb*, and *UAS-pros RNAi* (#26745; Bloomington stock) for rescuing Btd loss-of-function phenotypes.

### RNAi knockdown and clonal analyses

For RNAi knockdown analyses of Btd and Pros, larvae were raised at 29°C to increase the expression of *UAS-RNAi* transgenes after hatching. Furthermore, *UAS-Dcr2* was expressed together with

*UAS-RNAi* transgenes to enhance the efficiency of RNAi knockdown. For clonal analyses, MARCM (*Lee and Luo, 1999*) clones were induced by 1 hr heat shock at 38°C for 1 day after larval hatching. Larval brains were dissected at third instar larval stages for the examination of phenotypes.

## Immunostaining and confocal microscopy

Larval brains were dissected, fixed, and stained as described before (*Lee and Luo, 1999*). Primary antibodies used in this study include: rabbit anti-Mira (1:500), guinea pig anti-Ase (1:5000), rabbit anti-Dpn (1:500) (a gift from Y.N. Jan), rat anti-mCD8 (Life Technologies, Grand Island, New York, 1:100), mouse anti-Pros (Developmental Studies Hybridoma Bank, Iowa City, Iowa, 1:20), mouse monoclonal anti-α-tubulin (Sigma, St. Louis, Missouri, 1:1000), rabbit anti-dsRed (Clontech, Mountain View, California, 1:500), rabbit anti-PntP1 (1:500, a gift from JS Skeath). Secondary antibodies conjugated to Cy2, Cy3, Cy5, or DyLight 647 (Jackson ImmunoResearch, West Grove, Pennsylvania) were used at 1:100, 1:500, or 1:500, respectively. Images were taken with a Zeiss LSM510 confocal microscopy and processed with Adobe Photoshop. For quantifications of the number of mature INPs or the percentage of type II NB lineages with mature INPs, we focus on the medial group of type II NB lineages (lineages DM1–DM6). Two-tailed t-tests were used for statistics analyses.

## Acknowledgements

We thank RS Mann, YN Jan, JS Skeath, CY Lee, C Han, the Bloomington stock center, and the Developmental Studies Hybridoma Bank for fly stocks and antibodies; F Pignoni, ME Zuber, AS Viczian for thoughtful discussion and comments; F Pignoni for sharing research facilities. This work was supported by March of Dimes Basil O'Connor Starter Scholar Research Award (#5-FY14-59, S.Z), the National Institute of Neurological Disorders and Stroke of the National Institutes of Health under Award Number R01NS085232 (S.Z), and SUNY Upstate Medical University startup fund (S.Z).

## Additional information

### Funding

| Funder | Grant reference number | Author |
|---|---|---|
| State University of New York | Upstate Medical School | Sijun Zhu |
| National Institute of Neurological Disorders and Stroke | R01NS085232 | Sijun Zhu |
| March of Dimes Foundation | #5-FY14-59 | Sijun Zhu |

The funders had no role in study design, data collection and interpretation, or the decision to submit the work for publication.

### Author contributions

YX, SZ, Conception and design, Acquisition of data, Analysis and interpretation of data, Drafting or revising the article; XL, XZ, SM, HL, AU, Acquisition of data, Analysis and interpretation of data, Drafting or revising the article

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
