## [Decision Letter]

Thank you for sending your work entitled “The *Drosophila* Sp8 Transcription Factor Buttonhead Prevents Premature Differentiation of Intermediate Neural Progenitors” for consideration at *eLife*. Your article has been favorably evaluated by Janet Rossant (Senior editor), Marianne Bronner (Reviewing editor), and 2 reviewers.

The Reviewing editor and the reviewers discussed their comments before we reached this decision, and the Reviewing editor has assembled the following comments to help you prepare a revised submission.

*Drosophila* type-II neuroblast (NB) lineages produce neural cells indirectly, by generating transiently amplifying Intermediate Neural Progenitors (INPs). INPs play a crucial role in mammalian brain development and understanding how they form and proliferate is therefore a central question in neurobiology. In a previous study, the authors have shown that the Ets transcription factor Pointed P1 (PntP1) is required for the specification of Type-II NB lineages in *Drosophila* larval brains. PntP1 suppresses Asense (As) expression in NBs, which promotes the formation of INPs. Here, Xie et al. show that the product of the button head gene (btd) works cooperatively with PntP1 to generate functional INPs.

They first show that btd is required for the maturation of INPs: In its absence, INPs are still generated but they undergo premature cell cycle exit and differentiate into Ganglion Mother Cells (GMCs). Xie et al. elegantly demonstrate that this phenotype is due to the ectopic expression of Prospero (Pros) in the nuclei of newly generated INPs. Thus, btd prevents premature differentiation of INPs by repressing early Pros expression. The authors then test whether the btd phenotype might be due to the loss of PntP1. They show that although PntP1 expression decreases in the NBs of btd mutant type-II lineages (leading to ectopic Ase expression in a subset of these NBs), this defect is not at the origin of the loss of mature INPs. It is instead a secondary effect of the ectopic expression of Pros. These results nicely confirm the key role played by btd in the formation of mature INPs. Finally, Xie et al. show that co-expressing btd with PntP1 in type-I NB lineages is sufficient to convert most of them into type-II NB lineages. The major finding reported is that the transcription factor encoding btd gene is required in INPs to prevent their premature differentiation, and that this occurs because btd suppresses the (nuclear) expression of the homeodomain protein Prospero in INP sublineages. These interesting data add to our knowledge of the molecular mechanisms involved in the control of the (limited) proliferative potential of INPs in the *Drosophila* neural stem cell model. This paper is interesting, well written, and the data are convincing.

Major concerns that need to be addressed:

1) Some of the data presented has been published previously and are not declared as such. The data in Figure 1 in this paper correspond to that Figure 1 in previously published paper by the same senior author (53), with only artwork and magnification changed. The authors should check all figures carefully to ensure that this does not happen and not recycle the same data.

2) The authors should mention clearly that the use of btd-Gal4 to study Btd expression and drive transgenes might not faithfully reproduce the endogenous expression pattern of Btd. There are two reasons for this. First, enhancer-reporter constructs do not necessarily reflect endogenous protein expression. Moreover, if enhancer-reporter constructs do mimic endogenous expression in one tissue, this does not prove that the constructs mimic endogenous expression in other tissues. Clearly, immunolabelling or *in situ* hybridization would be preferable and should be added to the paper if an antibody is available. If this is not available, the authors should make sure that they explain that they have made all the controls to test whether the Gal4 line reproduces the expression pattern. In particular, the ability to rescue the loss of function phenotype with the human construct is a good argument that the Gal4 line is expressed in the right tissue. Rescue with the fly UAS construct should be added to confirm this.

3) Second, the btd-Gal4 construct used by the authors is an insertion in the *btd* gene that actually causes a mutation in the gene (Estella et al., 5929). This is not a serious issue since the authors always use it as heterozygous (the authors should confirm this in the text), but this fact should be explained and the authors should explain the controls that explain why this is not an issue.

4) In some quantifications, the authors have a sample size of n=3. They should quantify at least 6 or 7 samples for each experiment to further support their conclusions.

---

## [Author Response]

1) Some of the data presented has been published previously and are not declared as such. The data in Figure 1 in this paper correspond to that Figure 1 in previously published paper by the same senior author (53), with only artwork and magnification changed. The authors should check all figures carefully to ensure that this does not happen and not recycle the same data.

We apologize that we reused an image that has been published before without realizing it. We did not intentionally recycle the data, but we really appreciate that the reviewers noticed this problem. We should have looked more carefully to make sure the same image was not used again. We have replaced the reused image in Figure 1 with new ones with similar quality in the revision. We will also make sure that similar things do not happen again in the future.

2) The authors should mention clearly that the use of btd-Gal4 to study Btd expression and drive transgenes might not faithfully reproduce the endogenous expression pattern of Btd. There are two reasons for this. First, enhancer-reporter constructs do not necessarily reflect endogenous protein expression. Moreover, if enhancer-reporter constructs do mimic endogenous expression in one tissue, this does not prove that the constructs mimic endogenous expression in other tissues. Clearly, immunolabelling or in situ hybridization would be preferable and should be added to the paper if an antibody is available. If this is not available, the authors should make sure that they explain that they have made all the controls to test whether the Gal4 line reproduces the expression pattern. In particular, the ability to rescue the loss of function phenotype with the human construct is a good argument that the Gal4 line is expressed in the right tissue. Rescue with the fly UAS construct should be added to confirm this.

We agree with the reviewers that a GAL4 enhancer trap line like *btd-GAL4* does not always reflect endogenous expression patterns of the affected gene and it is essential to verify the *btd-GAL4* expression by immunostaining or *in situ* hybridization. However, we tried to generate btd antibodies using two different approaches (using a synthesized peptide and GST-btd fusion proteins), but had no luck. We also tried very hard to detect *btd* mRNAs by *in situ* hybridization. We got very nice in situ hybridization signals in the larval optic lobe (which is consistent with the expression of *btd-GAL4* in the optic lobe) and ventral imaginal discs, but we were not able to detect strong signals in the central brain or in the ventral nerve cord. Therefore, in order to test whether the *btd-GAL4* line reproduces the endogenous expression pattern, we tried to rescue the loss of function phenotype of Btd with the expression of mouse Sp8 construct (sorry we did not have the human Sp8 construct) or fly btd construct driven by *btd-GAL4* as reviewers suggested. Ideally this rescue should be done in *btd* mutant clones. However, we were not able to do such rescue experiments for the following two reasons:

(A) Although the *GAL4* insertion in the *btd-GAL4* line causes a lethal mutation of *btd*, we did not observe any obvious phenotypes in *btd-GAL4* homozygous mutant type II NB clones, indicating that the *GAL4* insertion does not affect the expression of *btd* in type II NB lineages (please see the new figure: Figure 6—figure supplement 1), which is possible given that the *GAL4* is inserted at 753bp upstream of the transcription start site of *btd* and the insertion of *GAL4* may not disrupt the enhancer elements that drive the expression of *btd* in type II NB lineages. The embryonic lethality of the *btd-GAL4* line could be due to the loss of Btd in other tissue or cells. Therefore, we could not do the rescue by using *btd-GAL4* to drive the expression of *UAS-mSp8* or *UAS-btd* in *btd-GAL4* homozygous mutant type II NB clones. In any event, we were able to rescue the lethality of *btd-GAL4* line by driving the expression of *UAS-btd* and type II NB lineages developed normally in those rescued *btd-GAL4* mutant larvae (data not shown, but we would be happy to provide images if needed).

(B) Since *btd-GAL4* is inserted in the promoter of *btd*, we could not do the rescue by using *btd-GAL4* to drive the expression of *UAS-mSp8* or *UAS-btd* in type II NB clones homozygous mutant for other *btd* mutant alleles, such as *btd*^*XG81*^.

Therefore, in order to rescue the loss of function phenotype of Btd with the expression of mouse Sp8 or fly Btd driven by *btd-GAL4*, we tried to rescue the Btd RNAi knockdown phenotypes in type II NB lineages by the expression of *UAS-mSp8* or *UAS-btd* driven by *btd-GAL4*. In a new supplemental figure (Figure 6—figure supplement 2), we showed that Btd RNAi knockdown driven by *btd-GAL4* completely eliminated mature INPs in nearly all type II NB lineages, which is much stronger than that of Btd RNAi knockdown driven by *pntP1-GAL4* (this could also indicate that the expression of *btd-GAL4* is likely more close to the endogenous expression pattern of Btd than *pntP1-GAL4* so that the knockdown of Btd driven by *btd-GAL4* is more efficient *pntP1-GAL4*, although we cannot rule out the possibility that the higher RNAi knockdown efficiency is due to the difference of the expression levels of *btd-GAL4* and *pntP1-GAL4*). This Btd RNAi knockdown phenotype could be fully rescued by the expression of *UAS-mSp8* driven by the same *btd-GAL4*. However, the expression of *UAS-btd* driven by *btd-GAL4* could only partially rescue the Btd RNAi knockdown phenotype. The incomplete rescue by *UAS-btd* is most likely because the *UAS-btd* construct contains the target sequence of *UAS-btd RNAi*. In any event, the rescue of the Btd RNAi knockdown phenotypes by the expression of *UAS-mSP8* and *UAS-btd* driven by *btd-GAL4,* together with strong *btd* mutant phenotypes in type II NB lineages (and the failure of PntP1 to induced INP-like cells in btd mutant type I NB lineages), strongly argue that *btd-GAL4* reflects the endogenous expression pattern of *btd*.

*3) Second, the btd-Gal4 construct used by the authors is an insertion in the* btd *gene that actually causes a mutation in the gene (Estella et al., 5929). This is not a serious issue since the authors always use it as heterozygous (the authors should confirm this in the text), but this fact should be explained and the authors should explain the controls that explain why this is not an issue*.

Thanks for reviewers’ suggestions. We now mentioned in the text that the insertion of GAL4 transgene in the *btd-GAL4* enhancer trap line causes a lethal mutation of *btd* and we only used *btd-GAL4* heterozygous female larvae for phenotypic analyses. We also explained in the text that the generation of INPs are not affected in *btd-GAL4* heterozygous mutant type II NB lineages as shown in Figure 6 or in *btd-GAL4* homozygous mutant type II NB clones as mentioned above (Figure 6—figure supplement 1). Therefore, *btd-GAL4* heterozygous mutant background would not affect the generation of INP-like cells and thus it should not cause any problems when the *btd-GAL4* line was used for our research.

4) In some quantifications, the authors have a sample size of n=3. They should quantify at least 6 or 7 samples for each experiment to further support their conclusions.

We have increased the sample size as the reviewers suggested and did the statistical analyses again based on new sample sizes. We also indicated in the figure legends the samples sizes are the number of brain lobes or the number of type II or type I NB lineages/clones.